# Shortest Path Algorithms for Pedestrian Navigation Systems

Kyriakos Koritsoglou [1,*], Georgios Tsoumanis [2], Vaios Patras [1] and Ioannis Fudos [1,*]

1 Department of Computer Science and Engineering, University of Ioannina, GR-45500 Ioannina, Greece; vpatras@gmail.com
2 Department of Informatics and Telecommunications, University of Ioannina, GR-47100 Arta, Greece; gtsoum@uoi.gr
* Correspondence: kkoritsoglou@uoi.gr (K.K.); fudos@uoi.gr (I.F.)

**Abstract:** Efficient shortest path algorithms are of key importance for routing and navigation systems. However, these applications are designed focusing on the requirements of motor vehicles, and therefore, finding paths in pedestrian sections of urban areas is not sufficiently supported. In addition, finding the shortest path is often not adequate for urban sidewalk routes, as users of these applications may also be interested in alternative routes that, although slightly longer, possess other desirable features and properties. According to the literature, the search for alternative routes is carried out mainly using the k-shortest paths (KSP) algorithm which represents an ordered list of all available alternatives. Even though various KSP algorithms have been proposed, to the best of our knowledge, there is no research addressing all issues inherent in a pedestrian navigation system. The purpose of this paper is to present a heuristic algorithm for graph datasets that implements a penalty-based method which, by increasing certain edge weights, effectively searches for the most accessible alternative paths in multi-route cases. To demonstrate how the algorithm works, we present experimental results on finding the most accessible paths in pedestrian sections of the historical center of Thessaloniki city.

**Keywords:** shortest path; k-shortest paths; navigation systems; pedestrian navigation; alternative routes

## 1. Introduction

Finding the shortest path between two locations on graph datasets that represent transportation networks is a core function of geographic information systems (GIS) applications that has been intensively studied over the last two decades. Dijkstra's algorithm [1] and its variants have thoroughly been used to solve the shortest path issue. However, for users navigating through pedestrian sections of urban areas, the calculation of the shortest route is not sufficient, because there are usually alternative routes that, although slightly longer, abide by other desirable preferences and restrictions (i.e., accessibility, safety, comfort, and convenience) and therefore are preferable to the shorter one.

Alternative routing, in the literature, has primarily been addressed using the k-shortest paths (KSP) algorithm [2,3] or by applying penalties to graph edges [4,5] to generate alternative graphs (AGs) to calculate alternative paths from them. However, alternative routes are computed entirely based on their similarity to the shortest path in these works, resulting in alternative paths that are quite similar to one another as they share many graph edges.

A significant parameter that should be considered is that routing applications that make use of the algorithms above have been implemented to compute routes mainly on road networks and not on pedestrian areas. A key difference between these two networks is that the former is mapped with directed graphs while the latter with undirected, which makes the calculation of alternative paths a more complex process [6]. In addition, navigating through pedestrian sections of urban areas is not an easy task for specific population groups (i.e., people using wheelchairs, elderly people with walking stick/cane, parents with baby

strollers, etc.), and therefore, before suggesting alternative paths, a GIS application must be able to assess the needs and priorities of those people, in terms of safety and accessibility [7].

Most routing algorithms to date employ only the actual distances of the urban sections registered in their databases, which correspond to edge weights of the graphs they traverse, and eventually return the routes having the lower total weight cost (total distance) [8]. As a result, existing algorithms should be modified to take into account additional factors other than the distance between two points on the map, so as to provide more personalized results that will better meet the user needs. If the level of accessibility of each pedestrian section is known in advance, this factor should be considered when calculating the shortest accessible paths instead of the absolute shortest paths. In addition, the total number of ramps contained in each route should also be considered as it indicates equal number of crossings, which require more effort, and therefore muscle strain, from wheelchair users. Hence, alternative routes that contain fewer crossings, even if they are longer, are usually preferable.

The motivation behind the proposed study was to improve accessibility level of people with mobility problems and especially wheelchair users. The daily movement of these people in the inhospitable centers of modern cities is a process with many challenges, especially when navigating in unknown or less familiar areas. In the pedestrian sections of urban areas, there are various obstacles (very narrow sidewalks, broken pavement tiles, stairs, bus stops, etc.) that restrict or possibly exclude the possibility of access for wheelchair users.

Therefore, the contribution of the present study is the implementation of a routing algorithm, exclusively for navigation in the pedestrian sections of urban areas, which will take into account the information on the level of pedestrian accessibility to ultimately route from the safest and most comfortable route, thus improving the level of mobility of specific population groups.

The purpose of this paper is to present the development of a heuristic algorithm, providing several improvements on the penalty-based alternative route calculation method by increasing certain edge weights depending on their level of accessibility or whether they represent a ramp, to effectively suggest the most accessible alternative paths for multi-route cases. We then present experimental results on finding the most accessible paths in pedestrian sections of the historical center of Thessaloniki city.

The remainder of the paper is organized as follows. Section 2 briefly presents previous related work on alternative path algorithms. Section 3 presents our proposed improvements for producing alternative graphs using the penalty-based method for alternative route calculation. Section 4 presents a thorough experimental evaluation of our proposed method, and finally, conclusions are offered in Section 5.

## 2. Related Work

In this section, various algorithmic approaches are analyzed, along with higher-level solutions and their disadvantages as mentioned in the literature. The first approach to be analyzed is the most common technique, the k-shortest paths computation between a source *s* and a target *t* [9] in order of increasing cost. As the weight of each edge is defined only by the actual distance between the nodes of the graph, some serious disadvantages occur. The computed alternative paths share many edges, which in many cases makes them difficult to be distinguished. Moreover, KSP algorithm does not take into account other preferences and characteristics, which can be important selection criterions for a GIS application user. One way to deal with this issue is assigning a very large value to k, but at the expense of a rather high computational cost, that can be prohibitive for real-time applications. Therefore, in the KSP method, due to the several similar paths that can occur, the candidate result set should be evaluated and further filtered with respect to a number of constraints such as the accessibility, their total length or the number of changes of pavement through ramps according to each use case scenario and subsequently determine the final result set [3].

In general, better results could be achieved under using penalty-based methodologies. In this way, the generated paths become dissimilar to the shortest path by adding a penalty on the weights of certain edges [10]. Each time the weight of some edges is updated, repeated shortest path queries are executed on the alternate graph, thus calculating the alternate paths using mostly Dijkstra's algorithm or a speedup variation of it. Then the shortest path edges are penalized, and a new query is executed. If the newly calculated shortest path meets the desired requirements and characteristics, it is added to the solution set. This process is repeated until a sufficient number of alternative paths is computed [11]. Akgün et al. [4] proposed a method which doubles the weight of each edge that lies on the shortest path. A similar method is used in [12], where the penalty is computed in terms of both the path overlap and the total turning cost. In [13] Schultes et al. propose a speedup technique for shortest path computation including edge weight changes. Finally, Jian Pu et al. propose a variation of the Dijkstra's shortest path algorithm using a logarithmic edge weight increment procedure [14].

The initial penalty value before each subsequent iteration is arbitrary and can result in poor performance, so this is a disadvantage for penalty-based methods that needs consideration and experimentation. On the one hand, high initial penalty values seem to result in different but, often, very long alternative routes. On the other hand, small penalty values require more iterations in order to compute the desired results. There are also cases where the calculated alternative routes are not satisfactory enough when compared with the initial shortest path.

Another work on alternative route problem for road networks that can be used for pedestrian routing is [8]. This work is focused on finding several reasonable routes and suggesting new ways to measure the quality of a solution of alternative routes by mathematical definitions based on the graph structure. In addition, several heuristic techniques are presented, such as Pareto optimality, Disjoint Paths and Plateau method for computing alternative routes as determining an optimal solution is NP-hard in general.

Moreover, in [9], a formal solution for the search of alternative paths problem in road networks is presented. The tested algorithms in this work are mostly under the concept of local optimality to find the best alternative paths; moreover, it is optimized and simplified enough for real-time applications. Therefore, the presented methodology takes into consideration various functions, such as fuel consumption, that can be transformed to more pedestrian variables, such as accessibility and safety. Although it seems that it is suitable to solve the pedestrian routing problem, the concept of local optimality does not work so well for short distance routing.

A different perspective on the routing problem can be found in [15] where a ranking system for the traditional computed routes is developed. This integrated solution uses governmental data, OpenStreetMap database and other similar web services. The main idea is to create a more personalized route suggestion based on users' individual preferences. Thus, the end user can dynamically change the contribution of the above sources to the overall ranking mechanism. At the same research work, a road scene complexity scoring mechanism is proposed that combines geospatial data, traffic, even sensors and Street view images as input to deep neural networks. The scoring mechanism estimates the perceived and the descriptive complexity of the road which can be used as an input for the routing systems to further filter and personalize their results. This more personalized approach is developed for driving circumstances, but theoretically, it can be used on pedestrian routing where the user needs may vary (i.e., wheelchair or walking stick users).

In contrast with the most solutions mentioned above, in this work the proposed method for finding accessible alternative paths accepts common edges with the shortest route as long as these edges are classified as accessible, because not all the sections of the pedestrian network have this classification. The main idea is to find alternative paths that have as many accessible edges as possible compared to the shortest one. Thus, if the accessibility ratio (number of accessible edges/total alternative path edges) is improved, then the alternative path is preferable as long as it does not exceed some experimentally

determined thresholds (for example, alternative path total length may not exceed 50% of the shortest path length).

In addition, the data of the pedestrian network representing the historic center of the city of Thessaloniki, collected for the purposes of this study, cannot be used by the algorithms referenced, as the graph dataset that represents road networks is different from the corresponding that represents the pedestrian networks. In the second case, each part of this network can be accessed in any direction from its starting point to its end one and vice versa. This does not happen in road networks that even if they are two-way must have a different traffic flow for each direction and therefore a different edge in the graph that represents them. This fact significantly increases the complexity in the case of pedestrian navigation. As for each starting point and destination, there is a large number of alternative routes, something that does not happen in road networks that the other referenced algorithms deal with.

Therefore, according to the data above, the results of the proposed algorithm cannot be compared with the other state-of-the-art algorithms described in the current section. In summary, the two most important features that are the advantages of the proposed method will be further emphasized. First, as mentioned in the summary of this study, this algorithm successfully addresses an issue of pedestrian navigation, based on the accessibility characteristics of the pedestrian network, that even the largest routing platforms have not been able to resolve effectively to date. Second, due to its smart design, the proposed algorithm has theoretically better performance in the process of calculating the alternative paths in relation to the algorithms mentioned in this section.

The following section describes in detail the principles and the innovation points of the proposed penalty-based algorithm for calculating accessible k-shortest paths.

### 3. Algorithm Description

In this section, the proposed algorithm and the parameters that affect its operation are described in detail. A pedestrian section network is a graph G = (V, E) consisting of an edge set E and a vertex set V that contains *n* vertices. Each edge e ∈ E is represented as an ordered pair of vertices, in the form "from vertex i to vertex j", denoted by e = (i,j), and it is associated with a calculated weight w(e), which in this work's use-case represents not only the actual distance between them but the resultant of some additional characteristics as well, found in urban sidewalks (e.g., accessibility level, if current section represents a crosswalk between two ramps, etc.).

A k-shortest path query, given two vertices *s, t* ∈ *V*, looks for *k* sequences of edges, that each one connects *s* to *t* so that the sum of the calculated weights of these routes is minimized. Let $P^k$ be the *k*th shortest path from *s* to *t*. Then,

$$p^k = \left[ u^k(1), \, u^k(2), \ldots, u^k(q_k) \right] \tag{1}$$

where $u^k(1)$ is *s* and $u^k(q_k)$ is *t*.

Note that the graph dataset for this study was collected from the mapping of pedestrian routes of the historic center of Thessaloniki city. A penalty-based strategy is used to find the shortest available alternative paths on nondirectional graphs G. In addition to the distance between the nodes linked, the edges of the graph include two more highly useful features for this purpose.

The first of them determines if the current edge represents a crosswalk between two ramps, allowing the number of crosswalks (*crosswalks_no*) contained in each alternative route to be known after a graph traversal. The second feature is related to the level of accessibility of each pedestrian section. For a wheelchair user to cross a portion of this network, this must be at least 1.5 m wide and free of impediments (e.g., trees, bus stops, stairs, etc.) for the wheelchair to move. If the above conditions are met, then the specific edge of the graph is characterized as accessible, and its level of accessibility is equal to 1 (*access_level* = 1).

Additionally, if a graph edge has one or more of the above-mentioned obstacles, or if its width is less than 1.5 m but more than 0.90 m (according to the UN accessibility directives for wheelchair users [16]), and it can be accessed even if it is difficult, the graph edge is characterized as less accessible, and its level of accessibility is equal to 4 (*access_level* = 4). Finally, if a segment is inaccessible, its accessibility level is set to 0 (*access_level* = 0), and the algorithm ignores it while calculating alternate routes.

According to United Nations' Convention on the Rights of Persons with Disabilities, a pedestrian section is considered accessible only when its width is more than 1.5 m, its surface is smooth and there are no obstacles into it that make the wheelchair movement difficult or completely inaccessible. However, there is a significant number of sidewalks where navigation, under certain conditions, is possible but clearly more difficult for people with limited mobility. These sections are classified as less accessible, but they are still included in the graph dataset as for some routes it is mandatory to pass through them [17].

Such cases are observed when in a single point or along the entire length of a section its width is less than 1.5 m but at least 0.9 m. Otherwise this section is considered inaccessible. This case concerns the movement in only one direction as the user cannot rotate the wheelchair, because a width of at least 1.5 m is required. A similar case arises when the surface of a section is not completely smooth (pebbles, broken pavement slabs, etc.) but does not prevent access to it. Finally, pedestrian sections that contain various floor elements creating elevation differences such as stairs or surface slopes (slopes of more than 10% are not accessible without the help of an attendant) are also characterized as less accessible.

Based on the above data, it is very clear how the pedestrian sections are categorized. Those that have been considered as accessible are not penalized, and the weight of their edges is equal to their actual length in meters. On the contrary, in those that have been characterized as less accessible, a penalty with a specific factor is applied in order to increase their edge weight and finally to be selected by the routing algorithm as part of an alternative route only when it is really necessary to traverse them.

The *access_level* parameter for the less accessible parts was set to 2 in the first version of the algorithm, but after the field measurements performed, it was discovered that a value of 4 produces more accurate results in each use case. This change was made to address the usual case in which one side of an urban block is less accessible than the other three and the algorithm must route the user of the application from point A to point B through the three accessible ones. As a result, after the penalty application procedure, the length of the least accessible side must be more than the sum of the three accessible sides. To satisfy this condition, the value of the *access_level* parameter was changed to 4.

The value of the penalty factor cannot be too high because on the one hand, it would completely exclude the less accessible edges of the graph, while there are cases where the wheelchair user has to pass through them, and on the other hand, the algorithm would suggest routes which in many cases are much longer than the shortest one. In the following section, a comparison of the operation of the algorithm with a value of the parameter *access_level* equal to 2 and with a value of *access_level* equal to 4 is made in the same areas that were used as use cases during the presentation of the first version of this algorithm. Then the operation of the two versions of the algorithm will be presented in another 2 new use cases where the first version did not effectively produce the most accessible routes. This was the reason that led us to adjust the value of the *access_level* factor to 4.

The presented approach searches for the $k = 10$ shortest paths and selects the one that is the most accessible. As a result, in addition to the total cost of each alternate path, a separate total cost is calculated as well. The sum of all the products of the actual distance of each graph edge with the property *access_level* of the corresponding edge yields this additional total cost. In order to avoid the less accessible edges as much as possible in the proposed routes, the above total effectively weights them in relation to the accessible ones. At the same time, the total number of crosswalks included in each alternate path is calculated to be evaluated later during the next steps of the algorithm.

The following step is to calculate the alternative paths, with the most accessible being the one that is returned to the user as a result. Once this search is completed, the average path length (*avg_path_length*) is determined as the average cost of all alternative routes. The average path length increased by an average edge length (*avg_edge_length*) is the threshold (*ap_length_threshold*) for accepting alternative paths of a total weight less than that value (i.e., *ap_length_threshold = avg_path_length + avg_edge_length*). This limit was set because wheelchair users are unable to cover long distances within urban areas, especially without the assistance of an attendant.

In order to calculate paths with as many non-overlapping edges as possible, the approaches of the penalty method for calculating the k-shortest paths mentioned in Section 2 penalize the weights of the shortest path after the end of the first iteration and successively the weights of every alternative path at the end of each iteration. The downside of these methods is that they must generate a separate alternative graph for each iteration, increasing their complexity and total execution time. In contrast to other approaches, the proposed k-shortest path penalty-based algorithm has been implemented in such a way that the weights of all graph edges are penalized by default before its execution rather than after each iteration if their accessibility level is equal to 4 or if a specific edge corresponds to a crosswalk.

The proposed implementation results in the generation of a single alternative graph AG, in which each edge weight is determined by multiplying its distance by its accessibility level (*edge_weight = edge_distance × access_level*), meaning that the less accessible parts' distance is substantially quadrupled. In addition, if a graph edge corresponds to a crosswalk, the average edge length (*avg_edge_length*) is added to its length. Using the generated alternative graph, the weights of the 10 alternative shortest paths found earlier are recalculated, so in the end, the route of the lowest total score is considered as the most accessible and returns as a result of the algorithm. The proposed algorithm is shown in detail in Algorithm 1 and in Flow Scheme 1.

---

**Algorithm 1** The Proposed Algorithm

---

1:    **procedure** PROPOSED-ALGORITHM
2:      G = (V, E) ▸ Creation of Graph G representing the pedestrian sections of C the study area
3:      *avgEdgeLength* = 0                  ▸ Average edge length initialization
4:      *edgeCounter* = 0                   ▸ Initialization of edges counter
5:      *source = source*
6:      *destination = dest*
7:      **for** $i \leftarrow 0$ to $E - 1$ **do**             ▸ For all nodes
8:         **for** $j \leftarrow 0$ to $e_{E(i)} - 1$ **do**       ▸ For all selected node's edges
9:            avgEdgeLength + = $e_j$
10:            *edgeCounter* + = 1
11:         **end for**
12:      **end for**
13:      *avgEdgeLength = avgEdgeLength/edgeCounter*
14:      *P = ShortestPaths (AG,source,dest,10)*     ▸ Calculate the 10 shortest paths
15:      *avgPathLength = 0*        ▸ Initialization of the average path length
16:      **for** $i \leftarrow 0$ to 9 **do**       ▸ For the 10 shortest paths calculated
17:         *avgPathLength + = P[i].length*
18:      **end for**
19:      *avgPathLength = avgPathLength/10*
20:      *apLengthThreshold = avgEdgeLength + avgPathLength*
21:      **for** $i \leftarrow 0$ to 9 **do**       ▸ For the 10 shortest paths calculated
22:         **if** *P[i].length > apLengthThreshold* **then**     ▸ If a path length exceeds the Threshold
23:            *P[i].remove* ▸ Remove current path from the result list
24:         **end if**

---

| Algorithm 1 The Proposed Algorithm | |
|---|---|
| 25:     **end for** | |
| 26:     *Rp* = 0 | ▸ Initialization of paths calculated weights |
| 27:     **for** $i \leftarrow 0$ to *P.size* **do** | |
| 28:         $Rp[i] = di \times li + ri \times avgEdgeLength$ | ▸ d: edge distance, l: access level, r: presence of ramp |
| 29:     **end for** | |
| 30:     **return** *min(Rp)* | ▸ Return the minimum calculated weight |
| 31: **end procedure** | |

Given the above, the *k* paths that are found in Equation (1) are then applied on the alternative graph *AG*. Let *d* be the edge distance and *l* be the access level. In addition, *r* will denote the presence of a ramp while $\bar{d}$ is the average edge length. Given that the initial *k* = 10 paths were calculated based on the actual distances *d* then the paths are re-calculated for *AG*,

$$R\left(P^k\right) = \sum_0^n d_n \times l_n + r_n \times \bar{d} \tag{2}$$

where $R(P^k)$ is the new calculated weight of the previously found $P^k$ path and *n* denotes the edges traveled when the path is the *k*th. Then, the purpose is to find the minimum $r \subset R$. In this sense,

$$r = \min\left(P^k\right) \tag{3}$$

The innovation of the proposed algorithm focuses mainly on two points. The first of these is the ability to search for alternative routes with criteria that best suit the profile and preferences of each user in addition to the total distance of a route. The alternative route search algorithms mentioned in Section 2 are based only on the actual distances of the sections, which correspond to the edge weights of the graphs they cross and ultimately return the routes with the lowest total weight cost. The proposed method manages to more effectively meet the needs of disabled people that face mobility problems.

The other algorithms based on the penalty method, after the calculation of each alternative path, increase the weights of the graph edges contained in it in order to exclude it from the next traversal of the graph, but any change in weights implies the creation of a new alternative graph, a process that requires computing resources. Therefore, to calculate *k* paths, an equal number of alternative graphs must be created. The second innovation point of the algorithm presented is the creation of only one alternative graph for any number of k parameter leading to significantly shorter calculation times for the output of the produced results.

The following section presents in detail the results of the proposed penalty-based algorithm and explains its operation in four different use cases within the area where the experiments were performed. In addition, in the last two use cases a comparison is made between the current and the first version of the algorithm that did not perform as expected in specific cases.

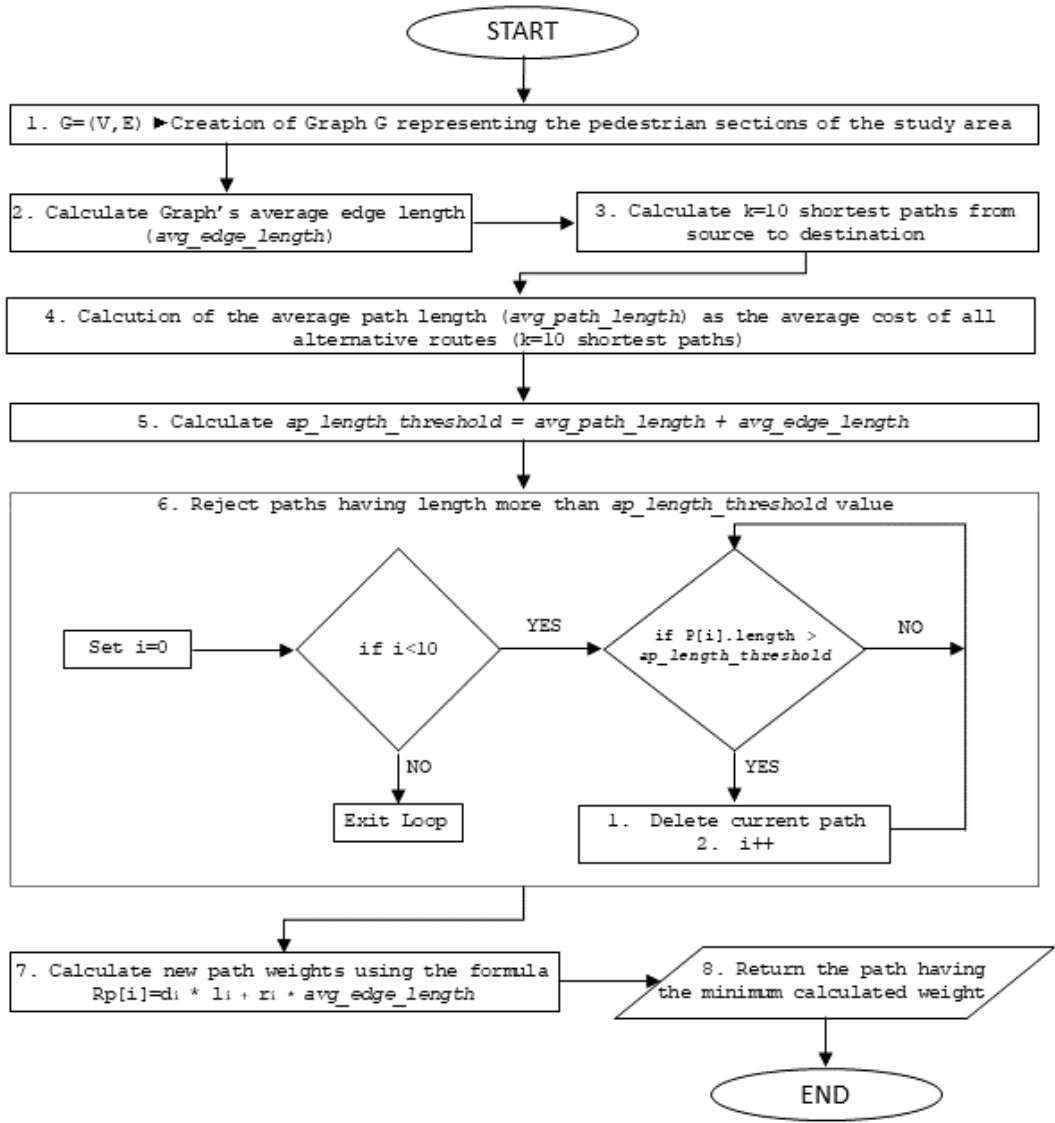

**Scheme 1.** Algorithm's Flowchart.

## 4. Research Results

The application of the proposed algorithm for the seach of the most accessible route in the historic center of the city of Thessaloniki gave, in most cases, very good results, two of which are presented in this section. However, there were specific cases when the starting point or destination was within a less accessible section, where the results were not as expected. That was the reason for the correction of the penalty factor. In this section, we will first present two use cases where both versions of the algorithm worked as expected and then two use cases in which the initial version of the algorithm fails to provide acceptable results.

In the following figures, the graph nodes are shown with red markers. Within each marker, its ID is displayed. The accessible edges are represented in green, while the less accessible ones are represented in yellow. Finally, the crosswalks are marked in blue. In each of the following use-cases, the graph of the specific area is initially presented on the Google maps web service, followed by the shortest route between the source and the target node, as well as the most accessible of the ten alternatives paths. In addition, the last two

use cases show the most accessible paths returned by both versions of the algorithm so that they can be compared. Finally, for each case, a corresponding table lists all the calculated weights of each of the k shortest routes. These tables show how the results shown in the figures below have been calculated.

*4.1. Use Case I*

In the first use-case (Figure 1), the transition from node 84 to node 245 is considered. As shown in Table 1, the shortest route passes through the following nodes: 84, 10, 9, 2, 80, 246, 254, 253, 252, 245 and has a total length of 353.3 m (Figure 2). However, we observe that a significant part of this route passes through sections that have been characterized as less accessible.

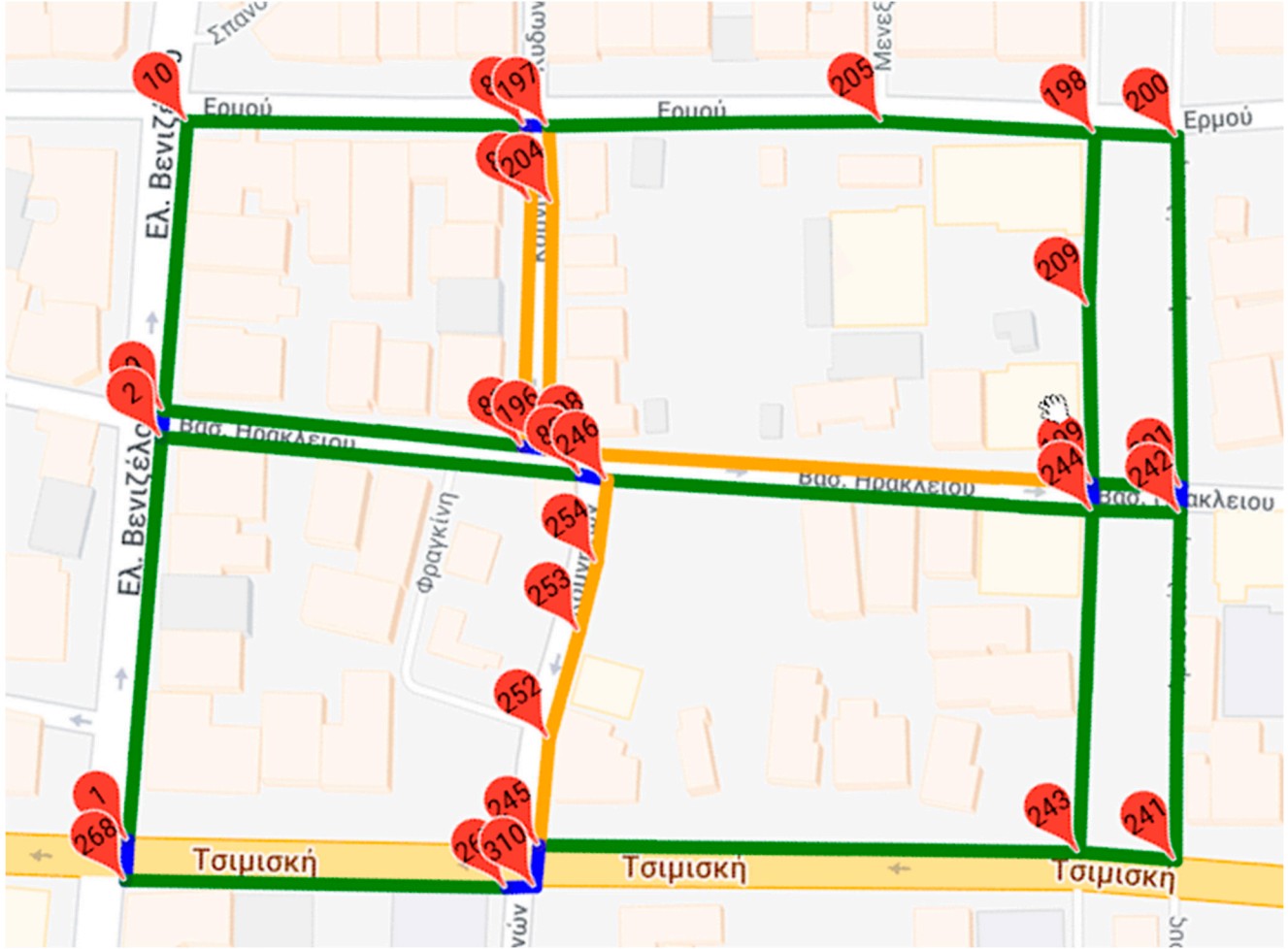

**Figure 1.** Use case I graph dataset.

When the weight of the specific route is recalculated in the alternative graph that has been produced, as described in the previous section, this will be equal to 518.4 m, while the route that crosses the nodes, 84, 197, 205, 198, 209, 199, 244, 243, 245, has a total weight of 514.50 m so it finally returns as the most accessible routes as shown in Table 1. In addition, the sum of the total weights of each route can be calculated using Table A1. We notice that in the second version of the algorithm, the most accessible path is the same as shown in the right part of Figure 2 but has a different weight due to the change in the penalty ratio. For a better understanding of the results in the following tables, it is pointed out that the shortest path appears with an orange background while the most accessible with light blue.

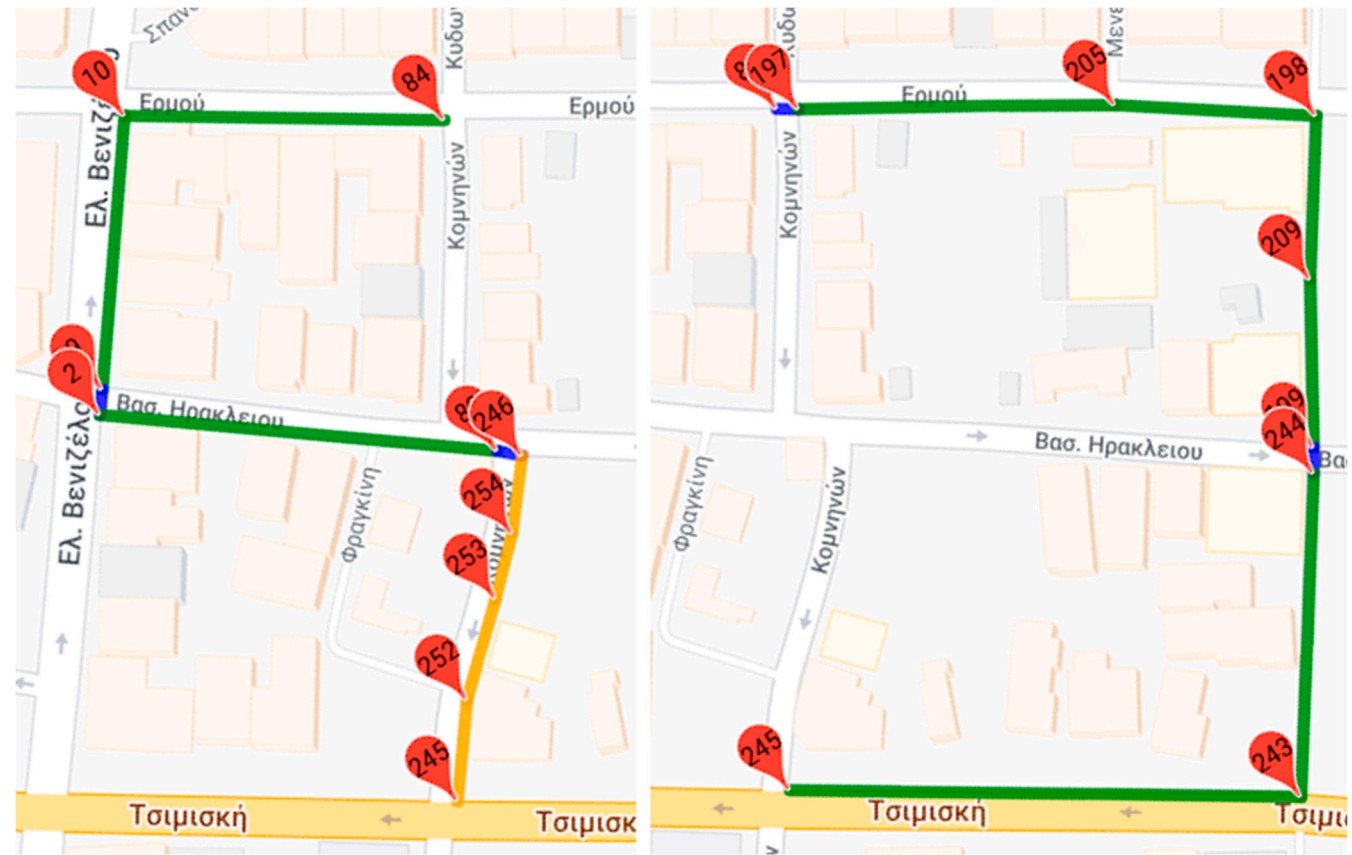

**Figure 2.** Shortest (**left**) and most accessible (**right**) path between nodes 84 and 245.

**Table 1.** k-shortest paths for use case I.

| Alternative Paths | Actual Distance | Crosswalk Counter | Calculated Distance V1 | Total Weight V1 | Calculated Distance V2 | Total Weight V2 |
|---|---|---|---|---|---|---|
| 84, 10, 9, 2, 80, 246, 254, 253, 252, 245 | 353.3 | 2 | 442.6 | 518.4 | 621.2 | 697.0 |
| 84, 10, 9, 2, 1, 268, 267, 310, 245 | 372.0 | 4 | 372.0 | 523.6 | 372.0 | 523.6 |
| 84, 85, 81, 9, 2, 1, 268, 267, 310, 245 | 385.2 | 4 | 462.3 | 613.9 | 616.5 | 768.1 |
| 84, 85, 81, 196, 208, 199, 244, 243, 245 | 432.0 | 2 | 640.2 | 716 | 1056.6 | 1132.4 |
| 84, 197, 204, 196, 208, 199, 244, 243, 245 | 432.4 | 2 | 641.2 | 717 | 1058.8 | 1134.6 |
| 84, 197, 205, 198, 209, 199, 244, 243, 245 | 438.7 | 2 | 438.7 | 514.5 | 438.7 | 514.5 |
| 84, 197, 205, 198, 200, 201, 242, 241, 243, 245 | 482.6 | 2 | 482.6 | 558.4 | 482.6 | 558.4 |
| 84, 10, 9, 81, 196, 208, 199, 244, 243, 245 | 591.6 | 2 | 722.7 | 798.5 | 984.9 | 1060.7 |
| 84, 10, 9, 2, 80, 246, 244, 243, 245 | 592.0 | 2 | 592.0 | 667.8 | 592.0 | 667.8 |
| 84, 85, 81, 9, 2, 80, 246, 244, 243, 245 | 605.2 | 2 | 682.3 | 758.1 | 836.5 | 912.3 |

The first column of Table 1 lists the sequences of nodes in each alternative path and the second column reports its actual distance. The third column contains the total crosswalk number of each route, and the fourth one indicates the distance resulting from the traversal of the alternative graph depending on the level of accessibility of each edge. The fifth column denotes the total weight of every route from which we determine which is the most accessible one. The fourth and fifth columns refer to the first version of the algorithm while the last two correspond to the same values in the current one.

### 4.2. Use Case II

In the second use case (Figure 3), the transition from node 258 to node 264 is examined. As shown in Table 2, the shortest route passes through the following nodes—258, 257, 260, 265, 288, 264—and has a total length of 218.93 m (Figure 4). Moreover, in this use-case, a significant part of this route passes through sections that have been characterized as less accessible. Moreover, the sum of the total weights of each path can be calculated using Table A2.

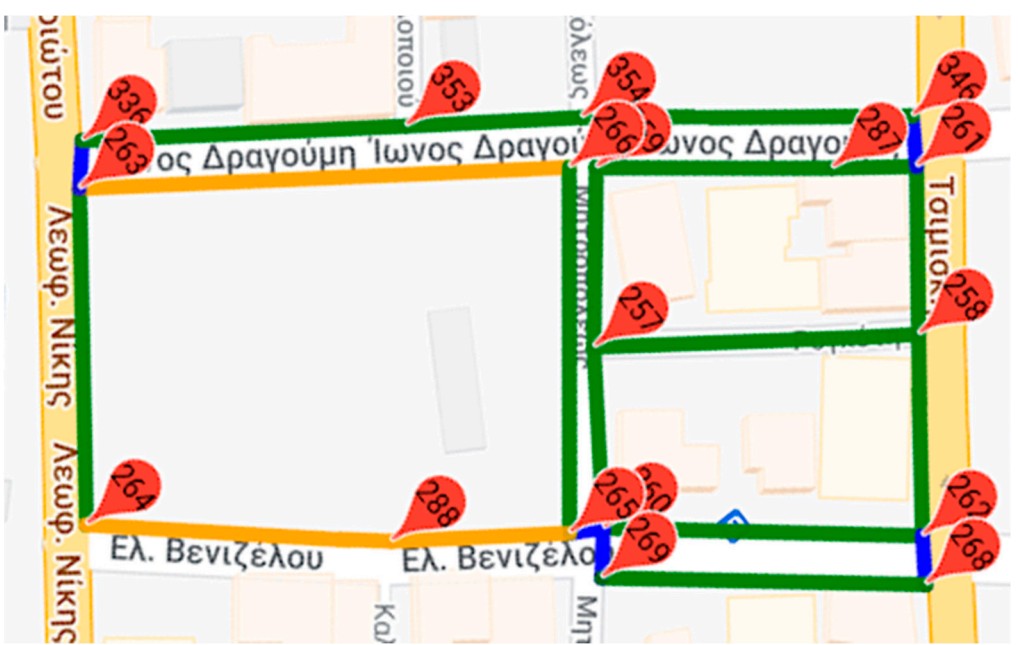

**Figure 3.** Use case II graph dataset.

**Table 2.** k-shortest paths for use case II.

| Alternative Paths | Actual Distance | Crosswalk Counter | Calculated Distance V1 | Total Weight V1 | Calculated Distance V2 | Total Weight V2 |
|---|---|---|---|---|---|---|
| 258, 257, 260, 265, 288, 264 | 218.9 | 1 | 322.6 | 360.5 | 530.0 | 567.9 |
| 258, 262, 260, 265, 288, 264 | 222.5 | 1 | 326.2 | 364.1 | 533.6 | 571.5 |
| 258, 262, 268, 269, 260, 265, 288, 264 | 244.2 | 3 | 347.9 | 461.6 | 555.3 | 669.0 |
| 258, 261, 287, 259, 257, 260, 265, 288, 264 | 292.7 | 1 | 396.4 | 434.3 | 603.8 | 641.7 |
| 258, 261, 346, 354, 353, 336, 263, 264 | 307.4 | 2 | 307.4 | 383.2 | 307.4 | 383.2 |
| 258, 257, 260, 265, 266, 263, 264 | 370.2 | 1 | 475.4 | 513.3 | 685.8 | 723.7 |
| 258, 262, 260, 265, 266, 263, 264 | 373.8 | 1 | 479.0 | 516.9 | 689.4 | 727.3 |
| 258, 262, 268, 269, 260, 265, 266, 263, 264 | 395.5 | 3 | 500.7 | 614.4 | 711.1 | 824.8 |
| 258, 261, 346, 354, 352, 351, 335, 336, 263, 264 | 417.2 | 2 | 417.2 | 493.0 | 417.2 | 493.0 |
| 258, 261, 346, 345, 352, 351, 335, 336, 263, 264 | 437.2 | 2 | 437.2 | 513.0 | 437.2 | 513.0 |

Respectively, the route that crosses the nodes 258, 261, 346, 354, 353, 336, 263, 264 as depicted in Figure 5 will have a total weight of 307.4 m so it eventually returns as the most accessible of the alternative routes. We notice again that in the second version of the algorithm, the most accessible path is the same.

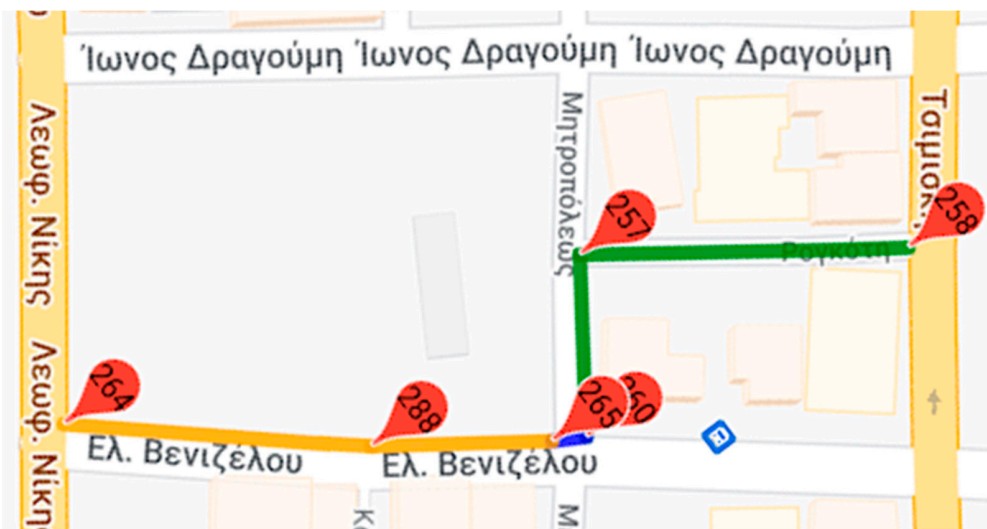

**Figure 4.** Shortest path between nodes 258 and 264.

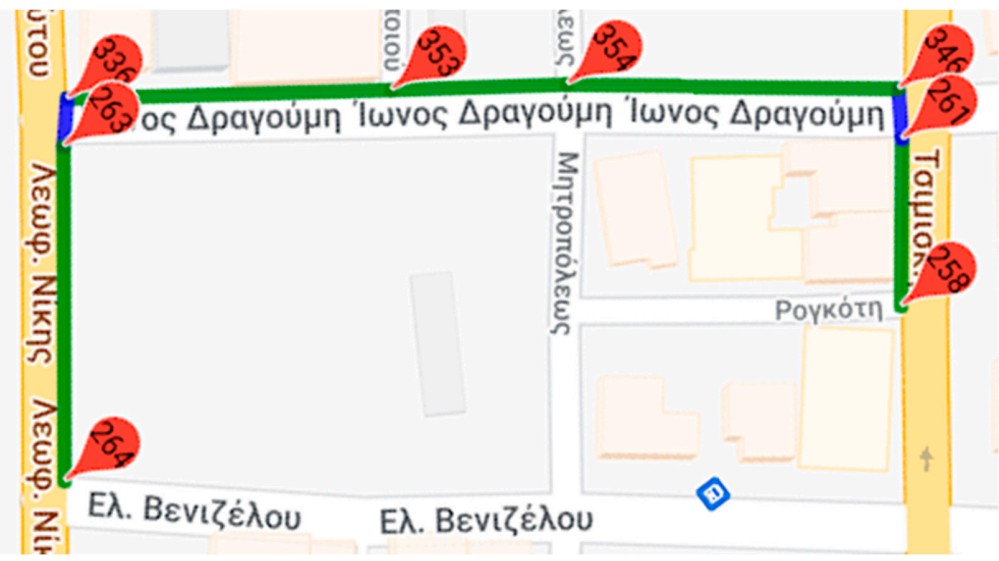

**Figure 5.** Most accessible path between nodes 258 and 264.

### 4.3. Use Case III

In the first two use cases, both versions of the algorithm calculated the same accessible paths as result. However, in the use cases that follow, the first version did not give the best results among the alternative ones, and therefore, after experimental tests on the penalty factor, its value was changed from 2 to 4. To make this difference clearer to the reader, an additional image was added, one for each version of the algorithm, which displays their results. In this use case (Figure 6), the transition from node 401 to node 446 is examined. As shown in Table 3, the shortest route passes through the following nodes—401, 400, 398, 405, 419, 424, 425, 426, 445, 446—and has a total length of 180.7 m (Figure 7).

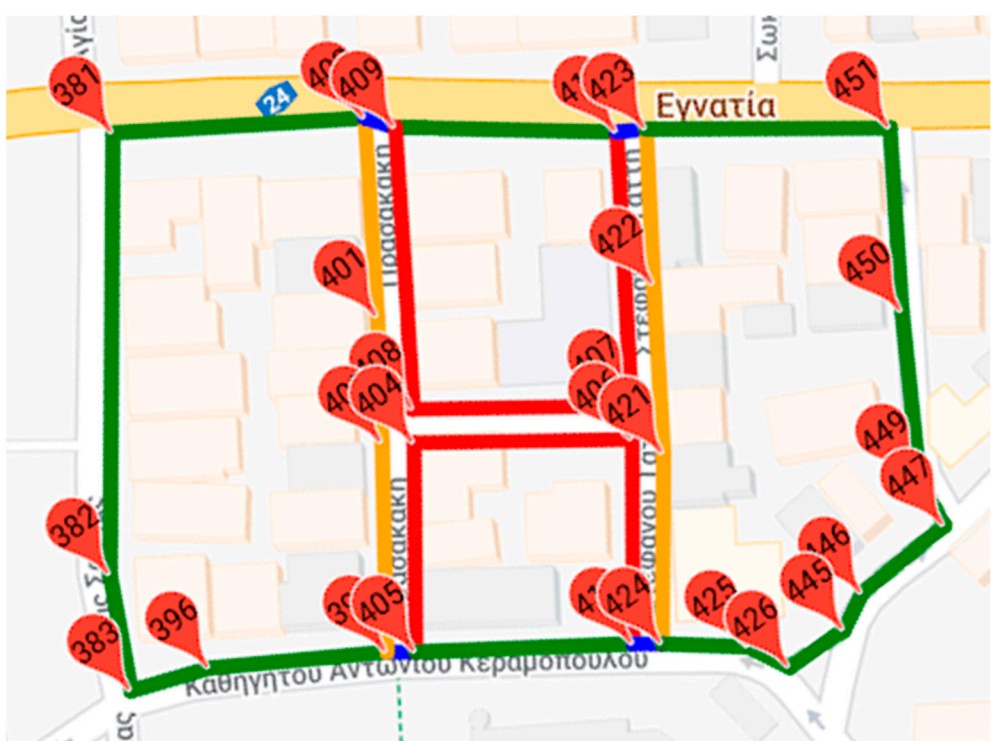

**Figure 6.** Use case III graph dataset.

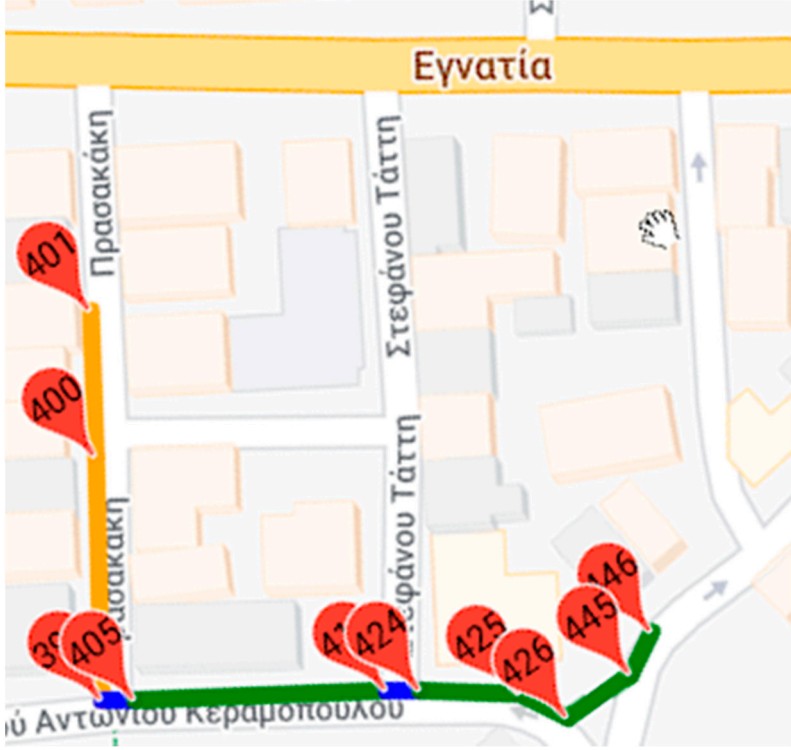

**Figure 7.** Shortest path between nodes 401 and 446 and also most accessible path between nodes 401.

**Table 3.** k-shortest paths for use case III.

| Alternative Paths | Actual Distance | Crosswalk Counter | Calculated Distance V1 | Total Weight V1 | Calculated Distance V2 | Total Weight V2 |
|---|---|---|---|---|---|---|
| 401, 400, 398, 405, 419, 424, 425, 426, 445, 446 | 180.7 | 2 | 252.1 | 327.9 | 394.9 | 470.7 |
| 401, 402, 409, 414, 423, 422, 421, 424, 425, 426, 445, 446 | 262.7 | 2 | 414.6 | 490.4 | 718.4 | 794.2 |
| 401, 402, 409, 414, 423, 451, 450, 449, 447, 446 | 263.0 | 2 | 305.3 | 381.1 | 389.9 | 465.7 |
| 401, 402, 381, 382, 383, 396, 398, 405, 419, 424, 425, 426, 445, 446 | 379.5 | 2 | 421.8 | 497.6 | 506.4 | 582.2 |
| 401, 400, 398, 405, 419, 424, 421, 422, 423, 451, 450, 449, 447, 446 | 400.2 | 2 | 581.2 | 657.0 | 943.2 | 1019.0 |
| 401, 402, 381, 380, 212, 213, 219, 220, 385, 384, 383, 396, 398, 405, 419, 424, 425, 426, 445, 446 | 462.5 | 6 | 504.8 | 732.2 | 589.4 | 816.8 |
| 401, 400, 398, 396, 383, 382, 381, 402, 409, 414, 423, 422, 421, 424, 425, 426, 445, 446 | 519.7 | 2 | 700.7 | 776.5 | 1062.7 | 1138.5 |
| 401, 400, 398, 396, 383, 382, 381, 402, 409, 414, 423, 451, 450, 449, 447, 446 | 520.0 | 2 | 591.4 | 667.2 | 734.2 | 810.0 |
| 401, 402, 381, 382, 383, 396, 398, 405, 419, 424, 421, 422, 423, 451, 450, 449, 447, 446 | 599.0 | 2 | 750.9 | 826.7 | 1054.7 | 1130.5 |

Because the starting point of the route is located on a part of the sidewalk that is characterized as less accessible (Figure 6), the preferred result is the one that minimizes the passage on it. Therefore, it should go from node 401 to 402 and then continue from the accessible sections of the pavement to the destination at node 446. However, in this case, for the first version of the algorithm, the most accessible path is the same as the shortest one. As shown in the first line of Table 3, the value of the actual distance field for this route is the lowest of the rest so it is the shortest path. Using Table A3 the sum of the total weights of each route can be calculated. In addition, when the weight of the specific route is recalculated using the first version of the alternative graph produced, this will be equal to 327.9 m (*Total Weight V1*). In this column, the value calculated is also the lowest, so this path is characterized as the most accessible.

After changing the penalty ratio, as shown in Figure 8, crossing the least accessible part is now the minimum possible. This fact is also reflected in the column *Total Weight V2* of Table 3 where we observe that in the third line the value of this field becomes the minimum, and therefore, the new most accessible route passes through the nodes 401, 402, 409, 414, 423, 451, 450, 449, 447, 446.

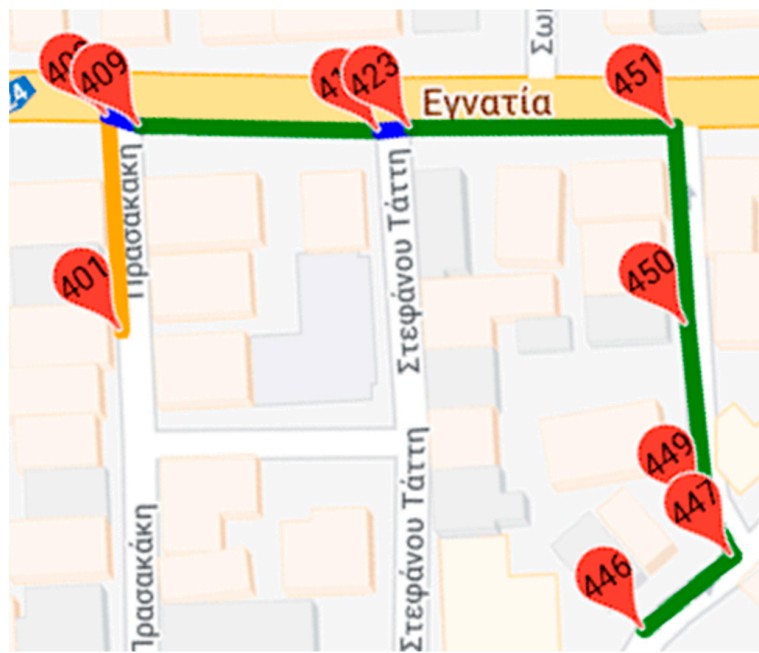

**Figure 8.** Most accessible path between nodes 401 and 446 using 2nd version of the algorithm.

### 4.4. Use Case IV

In the final use case (Figure 9) presented, the transition from node 458 to node 478 is examined. As shown in Table 4, the shortest route passes through the following nodes—458, 459, 470, 471, 479, 478—and has a total length of 165.8 m (Figure 10). In this use-case, a significant part of this route passes through sections that have been characterized as less accessible. Moreover, in this case, for the first version of the algorithm, the most accessible path is the same as the shortest one.

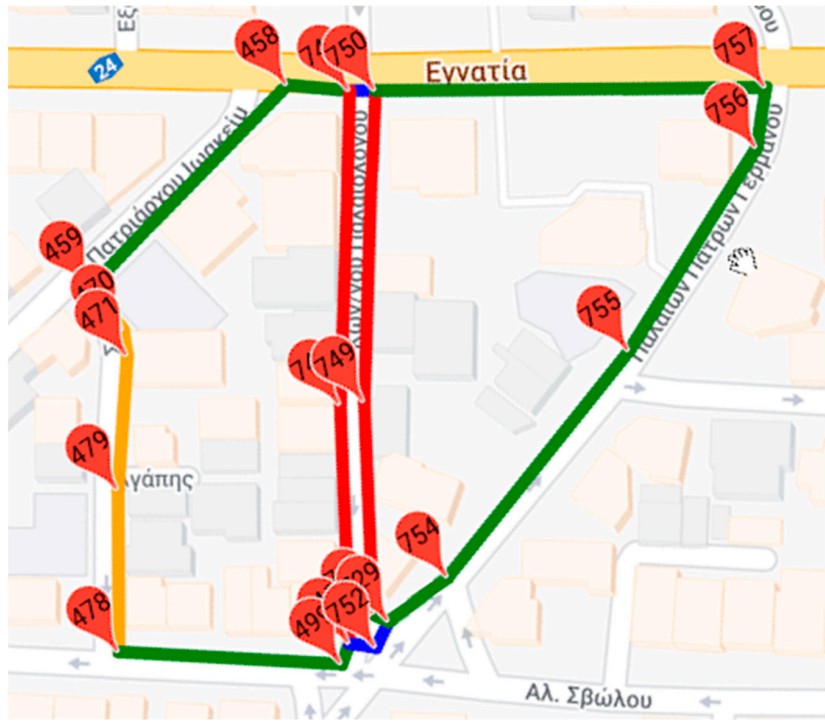

**Figure 9.** Use case IV graph dataset.

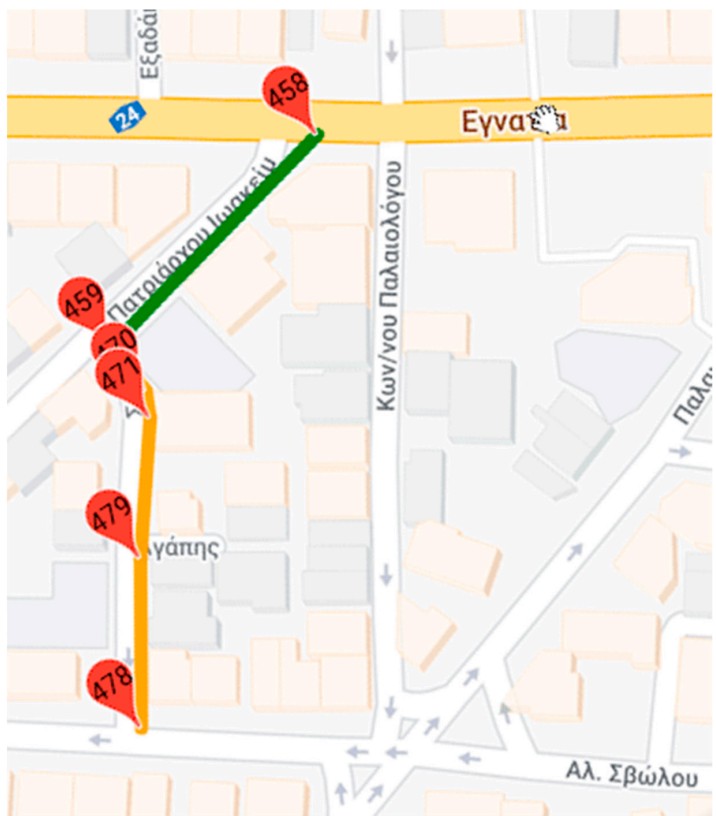

**Figure 10.** Shortest path between nodes 458 and 478 and the most accessible path between them using 1st version of the algorithm.

**Table 4.** k-shortest paths for use case IV.

| Alternative Paths | Actual Distance | Crosswalk Counter | Calculated Distance V1 | Total Weight V1 | Calculated Distance V2 | Total Weight V2 |
|---|---|---|---|---|---|---|
| 458, 459, 470, 471, 479, 478 | 165.8 | 0 | 260.8 | 260.8 | 450.8 | 450.8 |
| 458, 746, 750, 757, 756, 755, 754, 729, 752, 748, 499, 478 | 363.8 | 3 | 363.8 | 475.4 | 363.8 | 475.4 |
| 458, 746, 750, 757, 756, 755, 754, 729, 752, 514, 516, 517, 539, 518, 499, 478 | 415.8 | 6 | 415.8 | 639.0 | 415.8 | 639.0 |
| 458, 746, 750, 757, 756, 755, 754, 729, 752, 514, 515, 516, 517, 539, 518, 499, 478 | 433.5 | 6 | 433.5 | 656.7 | 433.5 | 656.7 |
| 458, 746, 750, 757, 777, 812, 807, 817, 810, 809, 526, 525, 524, 515, 514, 752, 748, 499, 478 | 546.3 | 6 | 546.3 | 769.5 | 546.3 | 769.5 |
| 458, 746, 750, 757, 777, 812, 807, 817, 810, 809, 526, 525, 524, 515, 516, 514, 752, 748, 499, 478 | 560.2 | 6 | 560.2 | 783.4 | 560.2 | 783.4 |
| 458, 746, 750, 757, 777, 812, 807, 817, 810, 809, 526, 525, 524, 515, 516, 517, 539, 518, 499, 478 | 568.0 | 7 | 568.0 | 828.4 | 568.0 | 828.4 |
| 458, 746, 750, 757, 777, 812, 807, 814, 818, 817, 810, 809, 526, 525, 524, 515, 514, 752, 748, 499, 478 | 581.1 | 6 | 581.1 | 804.3 | 581.1 | 804.3 |
| 458, 746, 750, 757, 777, 812, 816, 814, 807, 817, 810, 809, 526, 525, 524, 515, 514, 752, 748, 499, 478 | 581.4 | 6 | 581.4 | 804.6 | 581.4 | 804.6 |
| 458, 746, 750, 757, 777, 812, 816, 814, 818, 817, 810, 809, 526, 525, 524, 515, 514, 752, 748, 499, 478 | 581.8 | 6 | 581.8 | 805.0 | 581.8 | 805.0 |

As shown in the first line of Table 4, the value of the *actual distance* field for this route is the lower one. In addition, when the weight of the specific route is recalculated using the first version of the alternative graph that has been produced, this will be equal to 260.8 m (*Calculated Distance V1*). In this column, the value calculated is also the lowest, so this path is characterized as the most accessible. Using the new penalty ratio to calculate the weight of alternative routes, as the most accessible, as shown in Figure 11, is the one indicated in the second row of the Table 4.

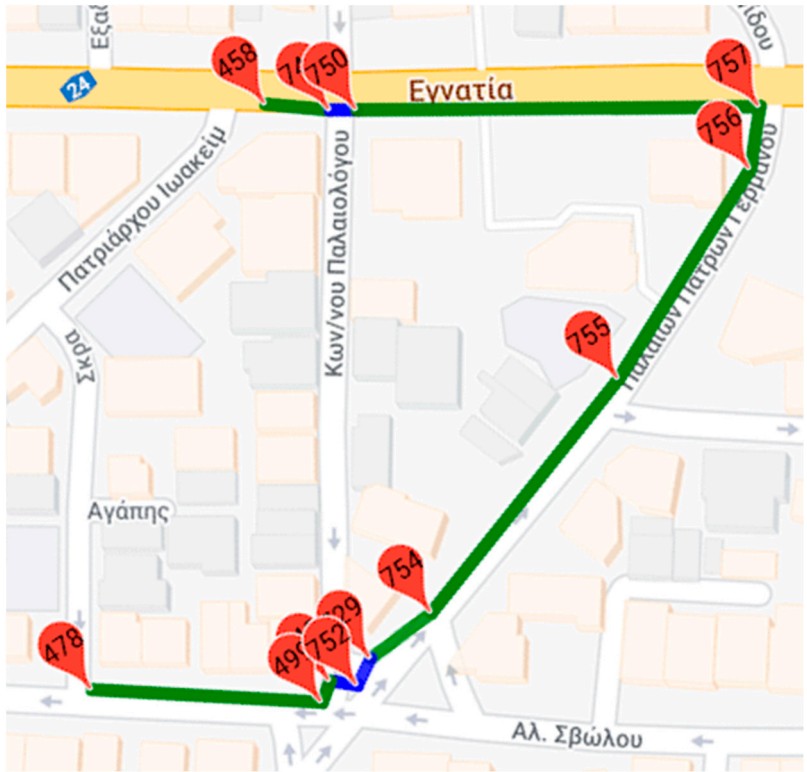

**Figure 11.** Most accessible path between nodes 458 and 478 using 2nd version of the algorithm.

This fact is also reflected in the column *Calculated Distance V2* of the same table where we observe that in the value of this field becomes the minimum and therefore the new most accessible route passes through the nodes 458, 746, 750, 757, 756, 755, 754, 729, 752, 748, 499, 478. This path, although longer, does not include any yellow sections (less accessible) unlike the first version of the algorithm, most of which was yellow. Therefore, with the change of the penalty ratio, in this case as well, the proposed algorithm behaves more efficiently. Using Table A4 the sum of the total weights of each path can be calculated.

Although the value of *Calculated Distance V2* of the second line of Table 4 is the minimum and the path depicted in Figure 11 consists of only accessible parts, this route, due to the fact that it crosses three ramps, does not have the lowest total weight (*Total Weight V2*) in relation to what the 1st version of the algorithm resulted. Therefore, the burden on the total weight of each alternative route due to the ramps enclosed in it should be studied and evaluated in subsequent versions of the proposed algorithm.

## 5. Conclusions

In this paper, we have proposed and described a penalty-based k shortest paths algorithm to search for the most accessible routes in the pedestrian sections of urban areas. The experimental tests that took place in the graph dataset representing pedestrian routes of the historic center of Thessaloniki city showed that the proposed algorithm returns, as a result, as the most accessible of the ten shortest paths between two nodes of this graph. To achieve this, the presented heuristic algorithm uses some additional features

found in the pedestrian sections apart from the actual distance (e.g., accessibility level, if current section represents a crosswalk between two ramps, etc.). The initial version of the algorithm proved effective in most cases. In order to work optimally in the whole urban pedestrian network, the penalty factor was corrected, and now the proposed algorithm manages to route through paths that pass through the minimum possible number of less accessible pedestrian sections. In a future version, the burden on the total weight of each alternative route due to the ramps enclosed in it will be studied. At the same time, the user will be allowed to choose if he wants the ramps to burden the overall result because, while it is a deterrent for wheelchair riders, the same does not happen for other categories of pedestrians.

**Author Contributions:** Conceptualization, K.K.; data curation, K.K.; methodology, K.K. and G.T.; software, V.P.; supervision, I.F.; writing—review and editing, K.K. All authors have read and agreed to the published version of the manuscript.

**Funding:** This research has been co-financed by the European Regional Development Fund of the European Union and Greek national funds through the Operational Program Competitiveness, Entrepreneurship and Innovation, under the call Research—Create—Innovate (project code: T1EDK-00108).

**Institutional Review Board Statement:** Not applicable.

**Conflicts of Interest:** The authors declare no conflict of interest.

**Appendix A**

In the figures presented in Section 4, the actual or calculated weights of every edge either of the initial or the alternate graph cannot be displayed due to constraints in the google maps platform API. For this purpose, four tables will be listed in this section, one for each use case, that will list the three properties (Actual Length, Accessibility Level, Is Ramp or Not) of every edge according to which the total and the total calculated route weights were calculated. The first feature corresponds to the actual length of the sidewalk between two nodes of the graph. The second identifies the level of accessibility of each segment, and the values it can receive are 1 or 2 for the first version of the algorithm and 1 or 4 for the second one. Finally, the third property gives the information whether an edge corresponds to a ramp or not.

Using this information, it will be possible to verify the results presented in Tables 1–4. That is, from the sequence of nodes of each route, it will be possible to calculate both the shortest and the most accessible path for each use case. For the calculation of the shortest path only, the first property (Actual Length) is used, while for the calculation of the most accessible, all three properties using the Equations (2) and (3) were found in Section 3.

**Table A1.** Use Case I.

| Source Node | Destination Node | Actual Length of Pedestrian Section | Is Ramp? | Accessibility Level of Pedestrian Section |
|---|---|---|---|---|
| 1 | 2 | 97.1 | 0 | 1 |
| 1 | 268 | 9.9 | 1 | 1 |
| 2 | 1 | 97.1 | 0 | 1 |
| 2 | 9 | 6.5 | 1 | 1 |
| 2 | 80 | 100.8 | 0 | 1 |
| 9 | 2 | 6.5 | 1 | 1 |
| 9 | 10 | 69.4 | 0 | 1 |
| 9 | 81 | 86.4 | 0 | 1 |
| 10 | 9 | 69.4 | 0 | 1 |

**Table A1.** *Cont.*

| Source Node | Destination Node | Actual Length of Pedestrian Section | Is Ramp? | Accessibility Level of Pedestrian Section |
|---|---|---|---|---|
| 10 | 84 | 80.9 | 0 | 1 |
| 80 | 2 | 100.8 | 0 | 1 |
| 80 | 246 | 6.4 | 1 | 1 |
| 81 | 9 | 86.4 | 0 | 1 |
| 81 | 85 | 58.9 | 0 | 2 |
| 81 | 196 | 5.9 | 1 | 1 |
| 84 | 10 | 80.9 | 0 | 1 |
| 84 | 85 | 18.2 | 0 | 2 |
| 84 | 197 | 5.7 | 1 | 1 |
| 85 | 81 | 58.9 | 0 | 2 |
| 85 | 84 | 18.2 | 0 | 2 |
| 196 | 81 | 5.9 | 1 | 1 |
| 196 | 204 | 59.1 | 0 | 2 |
| 196 | 208 | 10.8 | 0 | 2 |
| 197 | 84 | 5.7 | 1 | 1 |
| 197 | 204 | 18.6 | 0 | 2 |
| 197 | 205 | 79.9 | 0 | 1 |
| 198 | 200 | 19.7 | 0 | 1 |
| 198 | 205 | 51.3 | 0 | 1 |
| 198 | 209 | 41.3 | 0 | 1 |
| 199 | 201 | 21.2 | 0 | 1 |
| 199 | 208 | 120.3 | 0 | 2 |
| 199 | 209 | 42.6 | 0 | 1 |
| 199 | 244 | 6.6 | 1 | 1 |
| 200 | 198 | 19.7 | 0 | 1 |
| 200 | 201 | 83.7 | 0 | 1 |
| 201 | 199 | 21.2 | 0 | 1 |
| 201 | 200 | 83.7 | 0 | 1 |
| 201 | 242 | 6.6 | 1 | 1 |
| 204 | 196 | 59.1 | 0 | 2 |
| 204 | 197 | 18.6 | 0 | 2 |
| 205 | 197 | 79.9 | 0 | 1 |
| 205 | 198 | 51.3 | 0 | 1 |
| 208 | 196 | 10.8 | 0 | 2 |
| 208 | 199 | 120.3 | 0 | 2 |
| 209 | 198 | 41.3 | 0 | 1 |
| 209 | 199 | 42.6 | 0 | 1 |
| 241 | 242 | 83.1 | 0 | 1 |
| 241 | 243 | 22.9 | 0 | 1 |
| 242 | 201 | 6.6 | 1 | 1 |

**Table A1.** *Cont.*

| Source Node | Destination Node | Actual Length of Pedestrian Section | Is Ramp? | Accessibility Level of Pedestrian Section |
|---|---|---|---|---|
| 242 | 241 | 83.1 | 0 | 1 |
| 242 | 244 | 20.9 | 0 | 1 |
| 243 | 241 | 22.9 | 0 | 1 |
| 243 | 244 | 81.6 | 0 | 1 |
| 243 | 245 | 129.7 | 0 | 1 |
| 244 | 199 | 6.6 | 1 | 1 |
| 244 | 242 | 20.9 | 0 | 1 |
| 244 | 243 | 81.6 | 0 | 1 |
| 244 | 246 | 116.7 | 0 | 1 |
| 245 | 243 | 129.7 | 0 | 1 |
| 245 | 252 | 25.9 | 0 | 2 |
| 245 | 310 | 10 | 1 | 1 |
| 246 | 80 | 6.4 | 1 | 1 |
| 246 | 244 | 116.7 | 0 | 1 |
| 246 | 254 | 19.6 | 0 | 2 |
| 252 | 245 | 25.9 | 0 | 2 |
| 252 | 253 | 26.7 | 0 | 2 |
| 253 | 252 | 26.7 | 0 | 2 |
| 253 | 254 | 17.1 | 0 | 2 |
| 254 | 246 | 19.6 | 0 | 2 |
| 254 | 253 | 17.1 | 0 | 2 |
| 267 | 268 | 91.5 | 0 | 1 |
| 267 | 310 | 6.7 | 1 | 1 |
| 268 | 1 | 9.9 | 1 | 1 |
| 268 | 267 | 91.5 | 0 | 1 |
| 310 | 245 | 10 | 1 | 1 |
| 310 | 267 | 6.7 | 1 | 1 |

**Table A2.** Use Case II.

| Source Node | Destination Node | Actual Length of Pedestrian Section | Is Ramp? | Accessibility Level of Pedestrian Section |
|---|---|---|---|---|
| 257 | 258 | 69.7 | 0 | 1 |
| 257 | 259 | 38.6 | 0 | 1 |
| 257 | 260 | 38.5 | 0 | 1 |
| 258 | 257 | 69.7 | 0 | 1 |
| 258 | 261 | 36.1 | 0 | 1 |
| 258 | 262 | 43.1 | 0 | 1 |
| 259 | 257 | 38.6 | 0 | 1 |
| 259 | 287 | 50.6 | 0 | 1 |
| 260 | 257 | 38.5 | 0 | 1 |

**Table A2.** *Cont.*

| Source Node | Destination Node | Actual Length of Pedestrian Section | Is Ramp? | Accessibility Level of Pedestrian Section |
|---|---|---|---|---|
| 260 | 262 | 68.7 | 0 | 1 |
| 260 | 265 | 7 | 1 | 1 |
| 260 | 269 | 10.8 | 1 | 1 |
| 261 | 258 | 36.1 | 0 | 1 |
| 261 | 287 | 18.2 | 0 | 1 |
| 261 | 346 | 10.6 | 1 | 1 |
| 262 | 258 | 43.1 | 0 | 1 |
| 262 | 260 | 68.7 | 0 | 1 |
| 262 | 268 | 10.2 | 1 | 1 |
| 263 | 264 | 71.7 | 0 | 1 |
| 263 | 266 | 105.2 | 0 | 2 |
| 263 | 336 | 10.5 | 1 | 1 |
| 264 | 263 | 71.7 | 0 | 1 |
| 264 | 288 | 65.3 | 0 | 2 |
| 265 | 260 | 7 | 1 | 1 |
| 265 | 266 | 78.1 | 0 | 1 |
| 265 | 288 | 38.4 | 0 | 2 |
| 266 | 263 | 105.2 | 0 | 2 |
| 266 | 265 | 78.1 | 0 | 1 |
| 268 | 262 | 10.2 | 1 | 1 |
| 268 | 269 | 69.4 | 0 | 1 |
| 269 | 260 | 10.8 | 1 | 1 |
| 269 | 268 | 69.4 | 0 | 1 |
| 287 | 259 | 50.6 | 0 | 1 |
| 287 | 261 | 18.2 | 0 | 1 |
| 288 | 264 | 65.3 | 0 | 2 |
| 288 | 265 | 38.4 | 0 | 2 |
| 335 | 336 | 51.9 | 0 | 1 |
| 335 | 351 | 72.5 | 0 | 1 |
| 336 | 263 | 10.5 | 1 | 1 |
| 336 | 335 | 51.9 | 0 | 1 |
| 336 | 353 | 69.6 | 0 | 1 |
| 345 | 346 | 73 | 0 | 1 |
| 345 | 352 | 75.3 | 0 | 1 |
| 346 | 261 | 10.6 | 1 | 1 |
| 346 | 345 | 73 | 0 | 1 |
| 346 | 354 | 71.2 | 0 | 1 |
| 351 | 335 | 72.5 | 0 | 1 |
| 351 | 352 | 35.6 | 0 | 1 |
| 351 | 353 | 51.7 | 0 | 1 |

**Table A2.** *Cont.*

| Source Node | Destination Node | Actual Length of Pedestrian Section | Is Ramp? | Accessibility Level of Pedestrian Section |
|---|---|---|---|---|
| 352 | 345 | 75.3 | 0 | 1 |
| 352 | 351 | 35.6 | 0 | 1 |
| 352 | 354 | 57.1 | 0 | 1 |
| 353 | 336 | 69.6 | 0 | 1 |
| 353 | 351 | 51.7 | 0 | 1 |
| 353 | 354 | 37.7 | 0 | 1 |
| 354 | 346 | 71.2 | 0 | 1 |
| 354 | 352 | 57.1 | 0 | 1 |
| 354 | 353 | 37.7 | 0 | 1 |

**Table A3.** Use Case III.

| Source Node | Destination Node | Actual Length of Pedestrian Section | Is Ramp? | Accessibility Level of Pedestrian Section |
|---|---|---|---|---|
| 381 | 382 | 93.9 | 0 | 1 |
| 381 | 402 | 53.9 | 0 | 1 |
| 382 | 381 | 93.9 | 0 | 1 |
| 402 | 381 | 53.9 | 0 | 1 |
| 382 | 383 | 25.1 | 0 | 1 |
| 383 | 382 | 25.1 | 0 | 1 |
| 383 | 428 | 5.1 | 1 | 1 |
| 383 | 384 | 6 | 1 | 1 |
| 384 | 385 | 24.6 | 0 | 1 |
| 384 | 383 | 6 | 1 | 1 |
| 385 | 220 | 5.7 | 1 | 1 |
| 385 | 384 | 24.6 | 0 | 1 |
| 398 | 405 | 6 | 1 | 1 |
| 405 | 398 | 6 | 1 | 1 |
| 383 | 396 | 17.1 | 0 | 1 |
| 396 | 383 | 17.1 | 0 | 1 |
| 396 | 398 | 37.9 | 0 | 1 |
| 398 | 396 | 37.9 | 0 | 1 |
| 404 | 405 | 44.5 | 0 | 0 |
| 405 | 404 | 44.5 | 0 | 0 |
| 404 | 406 | 46.5 | 0 | 0 |
| 406 | 404 | 46.5 | 0 | 0 |
| 419 | 406 | 43.4 | 0 | 0 |
| 406 | 419 | 43.4 | 0 | 0 |
| 419 | 405 | 46.5 | 0 | 1 |
| 405 | 419 | 46.5 | 0 | 1 |
| 398 | 400 | 44.4 | 0 | 2 |
| 400 | 398 | 44.4 | 0 | 2 |
| 400 | 401 | 27 | 0 | 2 |
| 401 | 400 | 27 | 0 | 2 |

**Table A3.** *Cont.*

| Source Node | Destination Node | Actual Length of Pedestrian Section | Is Ramp? | Accessibility Level of Pedestrian Section |
|---|---|---|---|---|
| 401 | 402 | 42.3 | 0 | 2 |
| 402 | 401 | 42.3 | 0 | 2 |
| 402 | 409 | 6.7 | 1 | 1 |
| 409 | 402 | 6.7 | 1 | 1 |
| 409 | 414 | 47 | 0 | 1 |
| 414 | 409 | 47 | 0 | 1 |
| 408 | 409 | 60.2 | 0 | 0 |
| 409 | 408 | 60.2 | 0 | 0 |
| 407 | 408 | 46.5 | 0 | 0 |
| 408 | 407 | 46.5 | 0 | 0 |
| 407 | 414 | 58.4 | 0 | 0 |
| 414 | 407 | 58.4 | 0 | 0 |
| 414 | 423 | 6.4 | 1 | 1 |
| 423 | 414 | 6.4 | 1 | 1 |
| 422 | 423 | 32.9 | 0 | 2 |
| 423 | 422 | 32.9 | 0 | 2 |
| 421 | 422 | 36 | 0 | 2 |
| 422 | 421 | 36 | 0 | 2 |
| 421 | 424 | 40.7 | 0 | 2 |
| 424 | 421 | 40.7 | 0 | 2 |
| 419 | 424 | 6.1 | 1 | 1 |
| 424 | 419 | 6.1 | 1 | 1 |
| 423 | 451 | 51.8 | 0 | 1 |
| 451 | 423 | 51.8 | 0 | 1 |
| 450 | 451 | 38.8 | 0 | 1 |
| 451 | 450 | 38.8 | 0 | 1 |
| 449 | 450 | 36.2 | 0 | 1 |
| 450 | 449 | 36.2 | 0 | 1 |
| 447 | 449 | 11.1 | 0 | 1 |
| 449 | 447 | 11.1 | 0 | 1 |
| 446 | 447 | 22.7 | 0 | 1 |
| 447 | 446 | 22.7 | 0 | 1 |
| 445 | 446 | 8.9 | 0 | 1 |
| 446 | 445 | 8.9 | 0 | 1 |
| 445 | 426 | 13.7 | 0 | 1 |
| 426 | 445 | 13.7 | 0 | 1 |
| 425 | 426 | 9.5 | 0 | 1 |
| 426 | 425 | 9.5 | 0 | 1 |
| 424 | 425 | 18.6 | 0 | 1 |
| 425 | 424 | 18.6 | 0 | 1 |

**Table A4.** Use Case IV.

| Source Node | Destination Node | Actual Length of Pedestrian Section | Is Ramp? | Accessibility Level of Pedestrian Section |
|---|---|---|---|---|
| 458 | 459 | 70.8 | 0 | 1 |
| 459 | 458 | 70.8 | 0 | 1 |
| 459 | 470 | 13.7 | 0 | 2 |
| 470 | 459 | 13.7 | 0 | 2 |
| 470 | 471 | 6.6 | 0 | 2 |
| 471 | 470 | 6.6 | 0 | 2 |
| 471 | 479 | 33.8 | 0 | 2 |
| 479 | 471 | 33.8 | 0 | 2 |
| 478 | 479 | 40.9 | 0 | 2 |
| 479 | 478 | 40.9 | 0 | 2 |
| 478 | 499 | 56.2 | 0 | 1 |
| 499 | 478 | 56.2 | 0 | 1 |
| 746 | 458 | 15.9 | 0 | 1 |
| 458 | 746 | 15.9 | 0 | 1 |
| 746 | 750 | 6.3 | 1 | 1 |
| 750 | 746 | 6.3 | 1 | 1 |
| 750 | 749 | 78.5 | 0 | 0 |
| 749 | 750 | 78.5 | 0 | 0 |
| 746 | 747 | 78.9 | 0 | 0 |
| 747 | 746 | 78.9 | 0 | 0 |
| 747 | 748 | 60.2 | 0 | 0 |
| 748 | 747 | 60.2 | 0 | 0 |
| 748 | 499 | 5.4 | 0 | 1 |
| 499 | 748 | 5.4 | 0 | 1 |
| 749 | 751 | 53.9 | 0 | 0 |
| 751 | 749 | 53.9 | 0 | 0 |
| 729 | 752 | 7.2 | 1 | 1 |
| 752 | 729 | 7.2 | 1 | 1 |
| 729 | 751 | 4.3 | 0 | 1 |
| 751 | 729 | 4.3 | 0 | 1 |
| 748 | 752 | 6.9 | 1 | 1 |
| 752 | 748 | 6.9 | 1 | 1 |
| 750 | 757 | 98.5 | 0 | 1 |
| 757 | 750 | 98.5 | 0 | 1 |
| 756 | 757 | 15.2 | 0 | 1 |
| 757 | 756 | 15.2 | 0 | 1 |
| 755 | 756 | 60.4 | 0 | 1 |
| 756 | 755 | 60.4 | 0 | 1 |
| 754 | 755 | 72.9 | 0 | 1 |
| 755 | 754 | 72.9 | 0 | 1 |
| 754 | 729 | 18.9 | 0 | 1 |
| 729 | 754 | 18.9 | 0 | 1 |

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
