# Peer review of "Shortest Path Algorithms for Pedestrian Navigation Systems"

_information, doi:10.3390/info13060269_

Round 1
Reviewer 1 Report
In this paper, a shortest path algorithm for pedestrian navigation systems is proposed. This paper need a major revision:
(1) The motivation and contribution of this paper should be given point by point in introduction;
(2) In section 3, the innovation of the proposed algorithm should be emphysized. BTW, the figure of algorithm 1 is not clearly enough.
(3) It seems that there is no comparison in experimental part. The advantages of the proposed method should be compare with other state-of-the-art methods.
Author Response
Comment 1: “The motivation and contribution of this paper should be given point by point in introduction;”
Reply 1.1: Thank you for your comment. The following paragraph was added to the Introduction section:
The motivation behind the proposed study was to improve the accessibility level of people with mobility problems and especially wheelchair users. The daily movement of these people in the inhospitable centers of modern cities is a process with many challenges, especially when navigating in unknown or less familiar areas. In the pedestrian sections of urban areas there are various obstacles (very narrow sidewalks, broken pavement tiles, stairs, bus stops, etc.) that restrict or possibly exclude the possibility of access for wheelchair users. Therefore, the contribution of the present study is the design of a routing algorithm, exclusively for traffic in the pedestrian sections of urban areas, which will take into account the information on the level of pedestrian accessibility to ultimately route from the safest and most comfortable route contributing thus improving the level of mobility of specific population groups.
Comment 2: “In section 3, the innovation of the proposed algorithm should be emphysized. BTW, the figure of algorithm 1 is not clearly enough.”
Reply 1.2: The following paragraph was added in Section 3:
The innovation of the proposed algorithm focuses mainly on two points. The first of these is the ability to search for alternative routes with criteria that best suit the profile and preferences of each user in addition to the total distance of a route. The alternative route search algorithms mentioned in section 2 are based only on the actual distances of the sections, which correspond to the edge weights of the graphs they cross and ultimately return the routes with the lowest total weight cost. The proposed method manages to more effectively meet the needs of disabled people that face mobility problems.
The other algorithms based on the penalty method, after the calculation of each alternative path, increase the weights of the graph edges contained in it in order to exclude it from the next traversal of the graph. But any change in weights implies the creation of a new alternative graph, a process that requires computing resources. Therefore, to calculate k paths, an equal number of alternative graphs must be created. The second innovation point of the algorithm presented is the creation of only one alternative graph for any number of k parameter leading to significantly shorter calculation times for the output of the produced results.
We would also like to point out that due to scaling the figure of algorithm 1 is not clearly enough and for this reason it has been replaced with the following table to make it clearer.
Comment 3: “It seems that there is no comparison in experimental part. The advantages of the proposed method should be compare with other state-of-the-art methods.”
As mentioned in the introduction to the paper, to the very best of the authors’ knowledge, there is no routing algorithm in the literature based on the accessibility characteristics of the pedestrian network of urban areas for calculating the most accessible routes within it.
Reply 1.3: The following paragraph was added in section 2.
In addition, the data of the pedestrian network representing the historic center of the city of Thessaloniki, collected for the purposes of this study, can not be used by the algorithms referenced, as the graph dataset that represents road networks is different from the corresponding that represents the pedestrian networks. In the second case, each part of this network can be accessed in any direction from its starting point to its end one and vice versa. This does not happen in road networks that even if they are two-way must have a different traffic flow for each direction and therefore a different edge in the graph that represents them.
This fact significantly increases the complexity in the case of pedestrian navigation as for each starting point and destination there is a large number of alternative routes, something that does not happen in road networks that the other referenced algorithms deal with.
Therefore, according to the data above, the results of the proposed algorithm cannot be compared with the other state-of-the-art algorithms described in section 2. In summary, the two most important features that are the advantages of the proposed method will be further emphasized. First, as mentioned in the summary of this study, this algorithm successfully addresses an issue of pedestrian navigation, based on the accessibility characteristics of the pedestrian network, that even the largest routing platforms have not been able to resolve effectively to date. Second, due to its smart design, the proposed algorithm has theoretically better performance in the process of calculating the alternative paths in relation to the algorithms mentioned in this section. In section 3 there is a detailed description of this innovation point.

Reviewer 2 Report
The paper addresses an interesting topic that may be useful for defining the best path for pedestrian navigation systems.
Although the topic was well framed and the methodology used was well described, the paper needs to be improved in order to be published.
First, the question of what the authors understand by pedestrian accessibility and how it is determined deserves to be addressed in a separate section, and it should be kept clear how the classification of an accessible edge is made, ie, the criteria used and how they are translated in penalties.
Second, Algorithm 1 is an image and is difficult to read, so it should be improved. This Algorithm 1 must be presented in the form of a programming flowchart, not just programming code.
Third, it is at the level of results that the paper presents the greatest weaknesses. For each use case, the maps must have the same scale and cover the same area so that the reader can interpret and compare the results. Furthermore, the legibility of nodes and streets is poor, which makes it difficult to read them and interpret the results of the tables. This has to be improved.
The authors presented 4 use cases, but according to the reviewer what they presented is incomplete and does not allow the analysis of the conclusions obtained. It would be necessary to present a map for each use case with the classification value in terms of accessibility/penalty of the edges of this study area. With this map, it could be seen whether the results obtained would correspond to the value with the best accessibility indexes and closer to the values of the shortest path. The authors should also present maps with the ten paths, which would allow visualizing the differences between them, which would later be translated into the tables presented. It is important to note that the authors should also indicate the “most accessible alternative paths for multi-route cases” in each use case and on what bases it had been defined. In other words, there is a lack of a validation process for the results obtained.
The issue of correction of the “penalty factor” should be better explained.
Authors should create a discussion section where they should confront, or highlight the differences in the results of the method used concerning other methods listed in section 2 of this paper.
Author Response
"Please see the attachment."

Round 2
Reviewer 1 Report
After revision, this paper can be accepted now.
Author Response
Thank you for your comment.
Reviewer 2 Report
The authors answered the questions raised and significantly improved the way in which the results are presented, allowing a better analysis of them. Thus, the reviewer considers that the paper can advance in the review process.
However, the reviewer considers that the tables in Appendix A should be placed after the references section.
Author Response
Thank you for your comment.